# Volumetric-based Contact Point Detection for 7-DoF Grasping

**Junhao Cai**[1,2] **Jingcheng Su**[3] **Zida Zhou**[3] **Hui Cheng**[3] **Qifeng Chen**[1] **Michael Yu Wang**[2,4]

[1]The Hong Kong University of Science and Technology
[2]HKUST Shenzhen-Hong Kong Collaborative Innovation Research Institute, Futian, Shenzhen
[3]Sun Yat-sen University [4]Monash University

**Abstract:** In this paper, we propose a novel grasp pipeline based on contact point detection on the truncated signed distance function (TSDF) volume to achieve closed-loop 7-degree-of-freedom (7-DoF) grasping on cluttered environments. The key aspects of our method are that 1) the proposed pipeline exploits the TSDF volume in terms of multi-view fusion, contact-point sampling and evaluation, and collision checking, which provides reliable and collision-free 7-DoF gripper poses with real-time performance; 2) the contact-based pose representation effectively eliminates the ambiguity introduced by the normal-based methods, which provides a more precise and flexible solution. Extensive simulated and real-robot experiments demonstrate that the proposed pipeline can select more antipodal and stable grasp poses and outperforms normal-based baselines in terms of the grasp success rate in both simulated and physical scenarios. Code and data are available at https://github.com/caijunhao/vcpd

**Keywords:** Contact point detection, 7-DoF grasping, General object grasping

## 1 Introduction

Vision-based grasping is one of the most basic but important tasks in the robotics community. It is regarded as an operation primitive, and benefits a robot in handling more complicated manipulation applications [1, 2, 3]. Although this problem has been widely researched, it remains challenging, especially when the robot is confronted with general object grasping in cluttered environments [4, 5, 6, 7]. To deal with this problem, most previous works have tried to evaluate the 6-degree-of-freedom (6-DoF) pose (i.e., the position and orientation) of the end-effector given the vi-

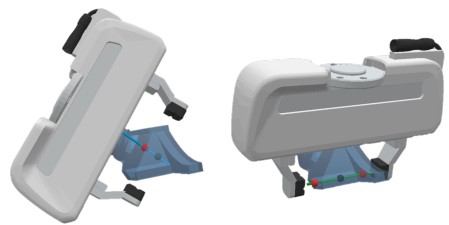

Figure 1: 6-DoF grasping based on left: surface normal and vertex or right: contact points.

sual observation of the scene [8, 9, 7, 10, 11, 12], which provides more flexible solutions compared with planar-based methods [13, 5, 14, 6]. However, most of the existing 6-DoF approaches suffer from three main limitations in terms of the pose representation and visual observation: 1) The gripper pose relies heavily on the assumptions that the end-point position of the gripper is estimated from a position located on the object surface (e.g., the red sphere on the left of Figure 1) and that the approach direction is parallel to the corresponding surface normal (e.g., the blue line in Figure 1) [8, 10, 11, 15, 16, 17, 18]. However, the first assumption may introduce position shift since most of the time the surface position is not exactly the same as the end-point position of the gripper (e.g., the black sphere on the left of Figure 1), which may result in failure when we empirically set the approach distance based on the surface position. Meanwhile, the second assumption may lead to no solution being found when all of the contact surface pairs in the normal direction are not parallel to each other. 2) Existing works mainly focus on estimating the gripper poses based on a single observation of the scene such as an RGB-D image or a point cloud [10, 19, 11, 20], which is visually insufficient, especially when tackling complex scenarios such as performing grasping in a

6th Conference on Robot Learning (CoRL 2022), Auckland, New Zealand.

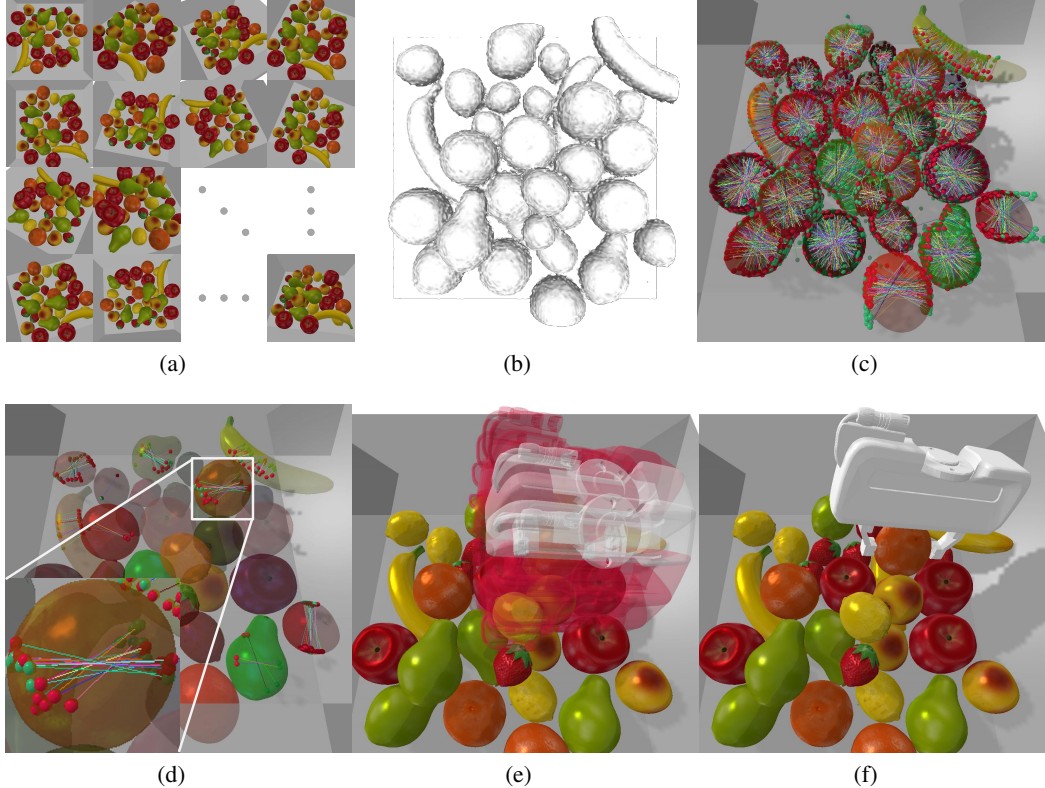

Figure 2: Overview of the grasp pipeline. (a) visualizes the views where the vision sensor captures the depth maps. (b) is the mesh generated from the TSDF volume. (c) visualizes the potential contact pairs sampled by the proposed sampling and matching heuristics. Red dots stand for the contact points generated by the Marching Cubes algorithm. The green dots denote matched contacts retrieved from the intersections between the grasp vectors and the isosurface of the TSDF volume. After the evaluation of grasp quality for each contact pair, the top 3‰ of high-quality pairs are selected, and collision checking is performed by rotating the gripper in the grasp direction on the TSDF volume, which are shown in (d) and (e), respectively. The final collision-free gripper pose will be obtained, as shown in (f).

cluttered environment. 3) Due to the partial observation, directly conducting collision checking is nontrivial. Although learning-based collision avoidance can be leveraged on the single-view data, large amounts of training data are required and the performance might degenerate when performing grasping on a different scene [10, 21, 22].

To resolve the limitations of current 6-DoF grasping methods, we propose a novel grasp pose detection method based on detecting contact point pairs in the truncated signed distance function (TSDF) volume to estimate both the 6-D pose and the width of the gripper, which we denote as 7-DoF grasping. Figure 2 illustrates the proposed pipeline. Our approach is closely related to the method proposed by Cai et al. in [12], while we circumvent the normal-based assumption and turn to detecting contact point pairs of the parallel-jaw gripper. The TSDF volume, a discretized representation of the scene where each voxel denotes a signed distance from current position to the closest object surface, allows multi-view fusion and provides more comprehensive shape information of objects that are essential for sampling and matching the contact point pairs. A contact point quality evaluation network is also proposed to predict the grasp quality of contact points in a pair-wise manner. Based on the evaluation results and the TSDF volume, post-processing operations, including volumetric collision checking and pose clustering, can be conducted efficiently to determine the approach vector [20] and width of the gripper alongside the contact point pair.

This novel grasping pipeline has three key characteristics: 1) The contact-based formulation, which directly provides the position of the grasp center and the grasp direction, can avoid the assumptions required by the normal-based methods [23, 24, 12]. 2) Instead of treating 3-D points of a single-

view point cloud as grasp contact candidates, this pipeline allows us to sample potential contacts from a TSDF volume, which provides more comprehensive scene observation and facilitates reliable sampling and matching of the contact point pairs. 3) Based on the TSDF volume and contact point evaluation network, collision-free poses with high grasp qualities can be obtained iteratively, guiding the robot to approach the target pose. Due to the eye-in-hand configuration, the vision sensor can capture different observations of the scene and thus enrich the information of the TSDF volume on the fly instead of multiple viewpoints being manually set beforehand [8, 23, 24, 25]. Therefore, the entire grasp pipeline can be executed in a closed-loop manner.

In summary, our main contributions are the following:

• A novel grasp pose estimation method based on contact point detection and evaluation on the TSDF volume.

• The contact point detection module can be integrated into the volumetric grasp pipeline, which allows collision-free 7-DoF closed-loop grasping.

• Extensive simulation and physical experiments demonstrate the superiority of the proposed contact-based method in terms of grasp success rate, antipodal score, and collision-free rate compared with normal-based ones [8, 23, 12].

## 2   Related Work

**Pose representation**. Many previous works have focused on resolving the grasping problem in a planar manner, which determines the gripper pose by estimating the position and the grasp angle of the gripper in the vertical direction [13, 26, 5, 14, 6]. However, this formulation inevitably restricts the ways objects could be grasped. In contrast, recent works have mainly favored the use of stereo data such as a point cloud or a depth map to predict 6-DoF gripper poses, which can provide more versatile solutions when handling complex scenarios [27, 7, 8, 11, 10, 23, 12, 17, 16, 18, 15]. Many of them use the surface normals of the objects as a prior of the approach direction when synthesizing grasp samples during data collection. Although this approach performs well in most cases, the performance might degenerate when there are no antipodal contact points constrained by surface normals. In addition, an ambiguity issue is also introduced by the normal-based methods, and researchers have resorted to performing angle discretization to eliminate the problem [12, 11].

Methods based on evaluating contact points have also been proposed [25, 20]. The contact-based methods directly regard the points of a point cloud as contact candidates and turn to estimating the grasp qualities of potential contact pairs. This formulation effectively circumvents the normal assumption, and directly obtains the end-point position of the gripper. However, existing contact-based methods attempt to find graspable contact pairs only on a partial view of the scene, which might be ill-posed since an antipodal contact point pair commonly exists on two opposite sides of an object, and only a single view of the scene might not include sufficient information.

**Scene observation**. Early works in vision-based grasping paid more attention to extracting shared graspable features for all objects from RGB images [13, 4, 26, 28]. Because of the explicit 3-D information of the scene and smaller gap for sim-to-real transfer, recent works have preferred to use depth maps or point cloud to conduct pose estimation [8, 14, 5, 17, 16, 6, 18]. However, these kinds of scene representations only reflect off the surface of the object, while an informative observation should naturally indicate both the geometric shape of each object and the spatial relationships between objects and the free space. Therefore, methods based on SDF volume have been proposed in which multiple views of depth map can be fused into a persistent 3D data structure, providing a more comprehensive representation of the scene that explicitly indicates both the occupied and free space [23, 24, 12].

## 3   Method

### 3.1   Problem Definition

In this work, we consider the clutter removal task where a robot tries to pick any object out from the workspace one by one when all the objects are stacked together. All the objects are unseen by the model, and we can only observe the scene through a vision sensor attached to an end-effector.

Suppose at time step $t$, we capture $t$ frames of depth maps $\mathcal{I}_t = \{I_1, ..., I_t\}$ with corresponding poses from the base to camera ${}_b\mathcal{T}_t^c = \{{}_bT_1^c, ..., {}_bT_t^c\}$ and the camera intrinsic matrix $K$. The TSDF volume can be computed by TSDF fusion [29], which is formulated as $V_t = f_{tsdf}(\mathcal{I}_t, {}_b\mathcal{T}_t^c, K)$. Our goal is to estimate the optimal contact pair $(p_t^*, p_t'^*)$ and its corresponding grasp vector $g_t^*$ and approach vector $a_t^*$ [20] from $V_t$. The pose and width of the gripper can be further computed from the quadri-tuple $(p_t^*, p_t'^*, g_t^*, a_t^*)$, denoted as

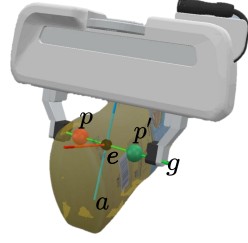

Figure 3: Pose representation of the gripper.

$$
\begin{aligned}
{}_bT_t^{g*} &= [{}_bR_t^{g*} | e_t^*], \\
{}_bR_t^{g*} &= [g_t^* \times a_t^*, g_t^*, a_t^*], \\
e_t^* &= (p_t^* + p_t'^*)/2,
\end{aligned}
\tag{1}
$$

where ${}_bR_t^{g*}$ denotes the rotation from the base to gripper frame, and $e_t^*$ is the end-point position of the gripper. The distance of two contact points can also determine the width of the gripper, which is computed as $||p_t^* - p_t'^*||_2$. An example of the gripper pose is shown in Figure 3.

To achieve this objective, a comprehensive volumetric-based grasp pipeline, including contact pair sampling and matching, grasp quality evaluation for contact pairs, and approach vector selection by collision checking, is proposed to perform the closed-loop grasping. An overview of the grasp pipeline is shown in Figure 2.

## 3.2 Contact Pair Sampling

For simplicity, a schematic illustration of contact point detection in 2-D space is shown in Figure 4. Given the current TSDF volume $V_t$, we execute the Marching Cubes algorithm [30] to retrieve the isosurface of the scene, which is represented as a mesh set including the surface vertices $\mathcal{P}_t = \{p_{1,t}, ..., p_{N,t}\}$ and their corresponding vertex normals $\mathcal{G}_t = \{g_{1,t}, ..., g_{N,t}\}$. The sampled vertices are considered as the potential contact points between one side of the fingertips and the objects, which are represented by the red dots in Figure 4. Moreover, the set of vertex normals determines the grasp direction of the gripper. We implement these settings based on the antipodal grasp rule [31], which says that to achieve a stable grasp, the grasp direction of a parallel-jaw gripper should be parallel to the surface normals of the contact points. With the above heuristic, we can further extrude all the grasp vectors in the inverse direction to obtain another set of contact points $\mathcal{P}_t' = \{p_{1,t}', ..., p_{N,t}'\}$ by finding the zero-value voxels in the TSDF volume. We neglect the samples that cannot find their matched con-

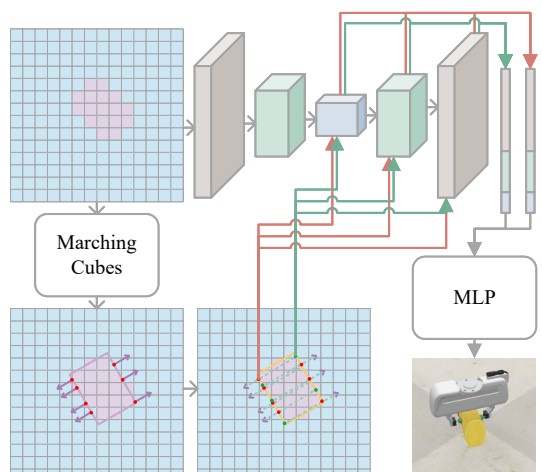

Figure 4: Illustration of contact point detection. Red dots and purple arrows are vertices and their surface normals respectively, as extracted by the Marching Cubes algorithm. Green dashed lines are the extrusion of the surface normals, and their intersections with the object are marked as green dots.

tact points in the inverse grasp direction or whose distances to their matched points are greater than the gripper width. For simplicity, we consistently use the symbol $N$ as the final number of sampled contact pairs in the subsequent sections.

Compared with existing works, our detection pipeline leads to 3 benefits: 1) The contact point pairs are sampled from a more comprehensive visual representation, which is more reliable than those sampled from a single point cloud. 2) The spatial information provided by the TSDF volume significantly simplifies the process of finding the matched contact points. 3) By making use of the surface normals, the sampled contact pairs are more prone to being antipodal, which is essential for stable grasping. Details about how to generate contact pairs in the TSDF volume will be illustrated in the appendix.

### 3.3 Contact Point Network

Based on a 3-D fully convolutional network (FCN) and multi-layer perceptron (MLP), we next propose a contact-point network (CPN) $f_\theta$ to evaluate the grasp quality pairwisely for all the contact pairs, which is also shown in Figure 4. We follow the network architecture proposed by [12] except that the last layer is replaced with a single-value output which ranges from $[0, 1]$, representing the grasp quality. The spatial features of the scene and the geometric features of the objects are first extracted by the 3-D FCN. We then use the sampled contact pairs as indices to retrieve the grasp features from the 3-D feature maps. Finally, the retrieved features for each contact pair will be sent to the MLP module to predict the corresponding grasp quality. The entire process can be formulated as $\widetilde{Q}_t = f_\theta(V_t, \mathcal{P}_t, \mathcal{P}'_t, \mathcal{G}_t)$, where $\widetilde{Q}_t \in \mathbb{R}^N$ represents the estimated grasp qualities for all the contact candidates. By leveraging the proposed CPN, we not only make the most of scene information provided by the TSDF volume but also circumvent redundant computation costs by only evaluating potential antipodal contact pairs.

### 3.4 Approach Vector Selection

So far, we have obtained the contact pairs with high grasp quality, which can determine the endpoint positions and the grasp vectors according to Eqn. 1. However, we cannot directly deduce the approach vectors from the contact pairs. Benefiting from the convenient collision checking in the TSDF volume, we can select the approach vector for each contact pair by rotating the gripper mesh on the axis of the grasp vector. One example is shown in Figure 2(e).

Assume that we have the gripper mesh model $m_g$ with its vertices denoted as $\mathcal{P}_g = \{p_{1,g}, ..., p_{N_g,g}\}$. For each contact pair $(p_{i,t}, p'_{i,t})$, we sample $N_a$ poses $_b\mathcal{T}^g_{i,t} = \{_bT^g_{1,i,t}, ..., _bT^g_{N_a,i,t}\}$ with their positions centered at the end-point position and the $y$ axes of their orientation aligned to the grasp vector. The $z$ axes, uniformly sampled in the subspace perpendicular to $y$, denote the potential approach vectors. Based on the TSDF volume, the vertex set, and the sampled poses of the gripper, we can check whether the gripper will collide with the objects by transforming the vertices with the sampled pose and evaluating the signed distance values retrieved by the transformed vertices in the TSDF volume. The poses whose retrieved signed distance values are all greater than $0$ (i.e., the transformed vertices are all located outside the objects) will be considered as feasible grasp poses. This operation can be denoted as $_b\widetilde{\mathcal{T}}^g_{i,t} = f_{cc\_sdf}(_b\mathcal{T}^g_{i,t}, \mathcal{P}_g, V_t)$, where $_b\widetilde{\mathcal{T}}^g_{i,t} \subseteq {}_b\mathcal{T}^g_{i,t}$ is the collision-free pose set for contact pair $i$. And the entire process can be summarized as $_b\widetilde{\mathcal{T}}^g_t = \{_b\widetilde{\mathcal{T}}^g_{1,t}, ..., _b\widetilde{\mathcal{T}}^g_{N,t}\}$, where $N$ is the number of graspable contact pairs and $_b\widetilde{\mathcal{T}}^g_t$ contains all the feasible poses. Finally, the final grasp pose $_bT^{g*}_t$ can be determined by performing non-maximum suppression (NMS) operation [12] on $_b\widetilde{\mathcal{T}}^g_t$.

## 4 Data Generation

To train the proposed CPN, we require training data consisting of the TSDF volume of the scene, sampled contact pairs, and their corresponding grasp qualities. Such data can be generated in three steps: grasp analysis on a single mesh, label validation on the simulator, and scene construction. Visualization of the synthesized data is available in the appendix.

**Grasp analysis on single mesh**. Before generating the cluttered scenarios with multiple objects stacked on top of each other, we first sample potential contact pairs on every single mesh and perform grasp analysis on all the candidates. Specifically, for each object mesh $m_i = \{\mathcal{P}_{m_i}, \mathcal{F}_{m_i}, \mathcal{N}_{m_i}\}$ represented as the set of vertices, faces, and vertex normals, we consider the vertices as contact points between one of the fingertips and the object. The opposite contacts $\mathcal{P}'_{m_i}$ for another fingertip are generated by finding the intersections between the object and the rays cast in the inverse normal direction. For each contact pair $(p_j, p'_j)$, we next evaluate the grasp quality by computing the antipodal score [12] and validating the collision status between the scene (the object $m_i$) and the gripper $m_g$, with its poses $_w\mathcal{T}^g_j$ determined by the contact points and the uniformly sampled approach vectors. If the antipodal score is greater than the given threshold and any of the poses conditioned on that contact pair lead to a collision-free grasp, we regard the candidate as graspable. Therefore, the output is

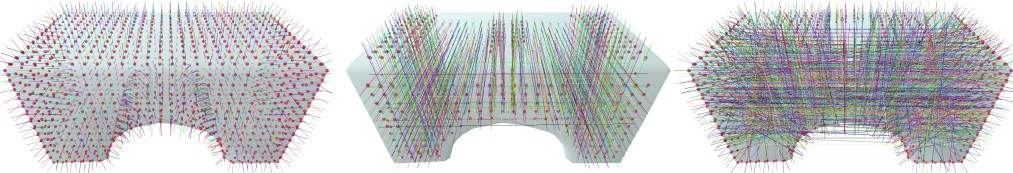

| (a) Contact points with normals | (b) Positive contact pairs | (c) Negative contact pairs |

Figure 5: Visualization of grasp analysis on a single mesh.

a binary vector $Q_{m_i} \in \{0, 1\}^{N_{m_i}}$, where each of the binary values is determined by

$$
q_j = \begin{cases} 1, & s_j \geq cos(\alpha_1) \cdot cos(\alpha_2) \ \ and \ \ f_{cc\_sim}({}_b\mathcal{T}_j^g, m_g, {}_b\mathcal{T}^{\mathcal{M}_i}, \mathcal{M}_i) \neq \varnothing, \\ 0, & otherwise \end{cases} , \qquad (2)
$$

where $s_j$ is the antipodal score for contact pair $(p_j, p'_j)$, $\alpha_1$ and $\alpha_2$ are the given thresholds of angles between the contact normal and the grasp direction, and $f_{cc\_sim}$ provides collision checking between the gripper $m_g$ and other meshes $\mathcal{M}_i$ given the gripper poses ${}_b\mathcal{T}_j^g$ and mesh poses ${}_b\mathcal{T}^{\mathcal{M}_i}$ respectively. In this step, $\mathcal{M}_i = \{m_i\}$. Similar to $f_{cc\_sdf}$ in Sec. 3.4, $f_{cc\_sim}$ outputs the set of collision-free poses in ${}_b\mathcal{T}_j^g$.

**Label validation**. Although analytical measures provide effective ways to generate grasp labels, they are not always reliable when such heuristics are transferred to the physical environment. Due to the rapid development of physics simulation, many works make use of a simulator to generate physically realistic grasp labels using a simulated robot arm to validate the graspability of the poses generated from analytical heuristics [15, 10, 32, 33]. In this work, we validate the grasp labels $\{Q_{m_i}\}$ by leveraging the Isaac Gym simulator [34], which can run thousands of environments in parallel with GPU acceleration. We follow the settings proposed by [33] to evaluate the reliability of all the candidate poses. One of the labeled meshes is shown in Figure 5.

**Scene construction**. Given sets of the mesh, contact pair, and grasp quality, we can build stacked scenes with labeled contact points. To achieve this, we first randomly pick and drop the meshes into a tray. The depth maps and their camera poses are respectively recorded as $\mathcal{I}_T$ and ${}_b\mathcal{T}_T^c$. All the contact pairs are then transferred to the scene according to the mesh poses, and collision checking is executed based on $f_{cc\_sim}$. Candidates that are both graspable and collision-free will be considered positive samples. Moreover, negative samples are collected based on three patterns, i.e., contact pairs 1) whose grasp qualities after label validation are 0, 2) whose grasp qualities after label validation are 1, while all the sampled poses on those contacts cause collisions in the scene, and 3) that are from positive samples but perform random permuting and re-matching, which generates a new set of contact point pairs. Finally, the depth maps $\mathcal{I}_T$ with corresponding camera poses ${}_b\mathcal{T}_T^c$ merge into the TSDF volume by TSDF fusion $V_t = f_{tsdf}(\mathcal{I}_t, {}_b\mathcal{T}_t^c, K)$, where $t$ ranges from 0 to $T$ such that we can save volumes that integrate different numbers of depth maps. Despite the multi-view observation, some areas containing contact points may be invisible. Therefore, we neglect samples which have signed distance values of the contact points larger than a specific threshold.

## 5    Experiments

This section presents extensive simulation and real-world experiments to evaluate the proposed grasp detection pipeline. We aim to validate 1) the superiority of the proposed contact-based method under the stacked scenarios compared with normal-based baselines and 2) the performance of the approach in a physical platform under different clutter removal scenarios. To this end, we compare the proposed method with three normal-based approaches, GPD [8], VGN [23], and VPN [12]. GPD performs grasp detection in a point cloud, while VGN and VPN detect grasp poses from the TSDF volume.

### 5.1    Evaluation Metrics

In this work, we use four metrics to evaluate the estimated grasp poses: 1) **antipodal score (AS)**: $s \geq cos(\alpha_1) \cdot cos(\alpha_2)$, where $\alpha_1$ and $\alpha_2$ are the angles of the grasp direction and contact normals

of the parallel-jaw gripper, 2) **collision-free rate (CFR)**: the percentage of grasp poses that do not cause collisions to grasp attempts on cluttered scenes, 3) **grasp success rate (GSR)**: the percentage of successful grasps compared to total grasp attempts, and 4) **grasp completion rate (GCR)**: the percentage of objects removed by the robot compared to the total number of grasp items. Following [12], we will manually remove any object that successively fails to be grasped three times.

## 5.2 Implementation Details

As demonstrated by Cai et al. [12], primitive-shaped objects can provide precise grasp annotation, and those grasp patterns are transferred well to unknown objects. We thus keep using the 191 primitive-shaped objects, including spheres, cubes, ellipsoids, cylinders, and building blocks released by [35] to generate the dataset. For scene construction, we follow the settings of VPN [12] to generate 8000 cluttered scenes with 5, 10, 15, and 20 objects stacked in the tray, respectively. For TSDF generation, four TSDF volumes are collected when 5, 10, 14, and 19 frames of depth images are integrated into the volume. All the training data are collected in PyBullet [36], and no real data are required.

During performance demonstration, primitive-shaped, procedural [37], and Kit [38] object sets are used for the AS and CFR evaluation in PyBullet and GSR evaluation in Isaac Gym. In the physical experiment, 32 unseen objects, comprising 8 fruits and 4 cups from the YCB object set [39], 8 household objects from YCB [39] and GraspNet [7], and 12 adversarial objects from DexNet [5] are used. Moreover, we use the Franka Emika robot arm with the RealSense D435 and a parallel-jaw gripper to perform grasp demonstrations. The vision sensor is mounted on the end-effector with its pose from the base to camera calibrated [40].

More details about the data, network architecture, parameter configuration, test objects, sampled cluttered scenes, and physical setting are posted in the appendix.

## 5.3 Simulation Experiments

We first investigate the antipodal scores and collision-free rates of the different methods by finding the contacts and collision status of the gripper given the pose estimated by the models in PyBullet. To demonstrate the robustness of the proposed methods, we respectively compute the average of the metrics on every 1000 scenes with 5, 10, 15, or 20 objects stacked together. We randomly render 20 depth maps from different viewpoints for each scene to generate the TSDF volume.

Table 1: AS / CFR / GSR

|  | # | VGN [23] | VPN [12] | VPN+GPRN [12] | CPN (Ours) |
|---|---|---|---|---|---|
| Primitive | 5 | 0.654 / 47.0 / 78.0 | 0.954 / 93.2 / 99.7 | 0.959 / 92.9 / 99.6 | **0.987 / 98.1 / 100.0** |
|  | 10 | 0.629 / 33.4 / 66.5 | 0.958 / 91.5 / **99.9** | 0.960 / 90.2 / 99.8 | **0.986 / 95.9 / 99.9** |
|  | 15 | 0.631 / 28.8 / 62.8 | 0.960 / 90.1 / 99.6 | 0.964 / 87.5 / 99.7 | **0.985 / 96.5 / 99.8** |
|  | 20 | 0.621 / 27.2 / 58.8 | 0.961 / 86.4 / 99.2 | 0.962 / 84.0 / 98.9 | **0.985 / 95.4 / 99.7** |
| Kit | 5 | 0.638 / 71.1 / 82.9 | 0.917 / 88.8 / 94.1 | 0.921 / 89.2 / 92.9 | **0.970 / 93.3 / 98.8** |
|  | 10 | 0.621 / 56.6 / 84.0 | 0.973 / 85.5 / 93.8 | 0.946 / 84.4 / 96.1 | **0.980 / 93.0 / 99.3** |
|  | 15 | 0.614 / 44.0 / 80.1 | 0.952 / 81.0 / 95.6 | 0.953 / 78.7 / 95.2 | **0.980 / 91.8 / 99.7** |
|  | 20 | 0.608 / 34.0 / 79.1 | 0.946 / 76.2 / 95.0 | 0.947 / 74.4 / 96.0 | **0.980 / 90.0 / 99.4** |
| Procedual | 5 | 0.424 / 47.9 / 71.6 | 0.801 / 69.3 / 92.9 | 0.830 / 69.3 / 93.0 | **0.945 / 89.6 / 98.4** |
|  | 10 | 0.470 / 51.3 / 68.1 | 0.794 / 67.2 / 90.7 | 0.818 / 66.9 / 92.2 | **0.939 / 88.2 / 98.6** |
|  | 15 | 0.483 / 43.4 / 63.5 | 0.796 / 66.0 / 90.1 | 0.814 / 66.7 / 89.8 | **0.940 / 88.6 / 98.7** |
|  | 20 | 0.471 / 38.6 / 65.9 | 0.785 / 62.9 / 89.4 | 0.801 / 62.0 / 88.6 | **0.942 / 90.3 / 98.1** |

The results, reported in Table 1, show that 1) the proposed contact-based method achieves state-of-the-art performance in terms of AS compared with the baselines, which implies that it can generate gripper poses that are more antipodal, and shows the superiority of the proposed contact-based method; 2) our method achieves similar performance on both the seen (primitive) and unseen (Kit and procedural) object sets, which demonstrates its generalization ability and robustness; and 3) our method outperforms the baselines on the CFR. We thus believe that the gripper widths estimated from the contact points and TSDF volume effectively decrease the risk of collision with other objects.

To further evaluate the reliability of the proposed method, we measure the GSR on all the object sets in the Isaac Gym simulator. The scene configuration and the number of depth maps are the same as in PyBullet. We randomly generate 1000 scenes with different levels of clutter and conduct one grasp attempt for each scene. The results, also reported in Table 1, demonstrate that our method can detect robust and reliable poses for the gripper on scenes with different clutter levels. More results and analyses of grasp experiments on other object sets are posted in the appendix.

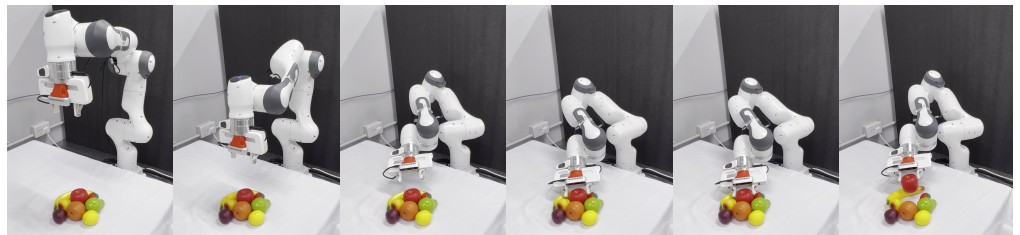

Figure 6: Demonstration of real robot grasping.

## 5.4 Real-robot Experiments

We design a series of clutter removal experiments on the four types of objects mentioned in Sec. 5.2. We select 8 (4 objects only for cup-shaped objects) objects of the same type for each experiment and randomly place them into the workspace to form a stacked scene.

Table 2: GSR / GCR

|  | Mugs ($4 \times 5$) | Fruits ($8 \times 5$) | Household ($8 \times 5$) | Adversarial ($8 \times 5$) |
|---|---|---|---|---|
| GPD [8] | 58.06 / 90.00 | 43.59 / 85.00 | 68.42 / 97.50 | 70.91 / 97.50 |
| VGN [23] | 80.00 / 100.0 | 56.25 / 90.00 | 39.74 / 77.50 | 41.56 / 80.00 |
| VPN [12] | **100.0 / 100.0** | 84.78 / 97.50 | 86.97 / **100.0** | 73.85 / 97.50 |
| VPN+GPRN [12] | **100.0 / 100.0** | 81.63 / **100.0** | 90.91 / **100.0** | 78.43 / **100.0** |
| CPN (Ours) | **100.0 / 100.0** | **93.02 / 100.0** | **95.24 / 100.0** | **85.11 / 100.0** |

$(m \times n)$ represents $m$ objects with $n$ test rounds.

We then execute the grasp according to the pose estimated by the models. We compare our method with GPD [8], VGN [23], and VPN [12]. Concretely, GPD estimates the gripper pose on a point cloud. For VGN, we follow the settings from [23] to generate the TSDF volume by moving the vision sensor to four viewpoints, while for VPN and our method, we keep using the VPN setting that the TSDF fusion and pose estimation are conducted on the fly, and the entire grasp trial is run in a closed-loop manner.

The GSR and GCR listed in Table 2 suggest that 1) our method achieves state-of-the-art performance on all types of grasp items, performing especially well on spherical objects, such as fruits, compared with the normal-based methods, which further demonstrates the ability to obtain antipodal grasp poses; 2) though trained with only primitive-shaped objects and synthesized images, the proposed pipeline can be directly applied to handling unknown objects in physical environments, which implies that the generated scenes are complex enough for the model to learn to detect graspable regions; and 3) although the viewpoints of the vision sensor are different from those set in the simulator during the TSDF fusion process, all the modules that require the TSDF volume still perform well, which shows the robustness of our method.

## 6 Conclusion and Limitations

This paper presents a novel vision-based grasp pipeline for detecting and evaluating contact pairs on the TSDF volume. The key advantages of this work are 1) the use of the TSDF volume, a comprehensive representation of the scene that allows reliable sampling of antipodal contact pairs, geometry-aware grasp quality inference, and collision checking with real-time performance, and 2) contact-based pose representation that effectively circumvents the ambiguity introduced by the normal-based approaches. Simulation experiments show that our method can efficiently avoid collision and compute more antipodal grasp poses. Clutter removal experiments show that our method trained with only synthesized data performs significantly better than other volumetric methods on all objects, which further demonstrates its ability of generalization.

Despite the advantages presented by the TSDF volume, this volumetric-based pipeline suffers limitations when handling slim- or flat-shaped objects in the physical environment. Due to the noise introduced by the consumer-level sensor and the truncated margin of the TSDF volume, the slim parts of the objects might be missing or the side surfaces of flat objects may be prone smoothing during the TSDF fusion, which leads to no contact points being found. To overcome this limitation, a more robust heuristic to infer the contact points and the contact normals will be considered in future work. More details about the failure cases are illustrated in the appendix.

Moreover, this work can only infer contact points for a parallel-jaw gripper. It would be a promising direction to investigate how to apply this contact-based pipeline for multi-finger grippers.

**Acknowledgments**

This work was supported in part by the Hetao Shenzhen-Hong Kong Science and Technology Innovation Cooperation Zone under the project HZQB-KCZYB-2020083. The authors would like to thank all the reviewers for their constructive feedback and for helping us make our paper more organized.

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
