# OpenReview forum: "Volumetric-based Contact Point Detection for 7-DoF Grasping"
_robot-learning.org/CoRL/2022/Conference — CoRL 2022 Poster_

### Official Review · Reviewer_EzWb · 2022-06-30

**Originality:** Good
**Technical Quality:** Very Good
**Clarity Of Presentation:** Very Good
**Impact:** 4

**Recommendation:**

Weak Accept: I recommend accepting the paper, but will not argue for my recommendation if the majority of other reviewers have a different opinion.

**Summary:**

The paper presents a 7-dof grasp point detection method. The proposed method considers the TSDF volume and trains a network to detect the grasp point using simulators. The paper validates the method by comparing the method with other methods using various objects in simulations and real robot experiments.

**Issues:**

All issues have been resolved in the revision.

**Quality Of The Limitations Section:**

Limitations are addressed clearly

**Reviewer Expertise:**

4: The reviewer is confident but not absolutely certain that the evaluation is correct

**Robotics Focus:**

Sufficient demonstration on hardware

**Strengths And Weaknesses:**

Strengths
- The introduction describes clearly the background and contribution of the paper.
- The related work section introduces related topics and papers and compares them with the proposed method.
- The method section, data section, and experiment sections clearly demonstrate the method, data, and experiments, respectively.
- The limitation part describes the limitation of the method precisely.

Weakness
- All issues have been resolved in the revision.

**Summary Of Recommendation:**

This reviewer recommends Weak Accept. The paper is written clearly and experimental results support the claim of the paper well. However, the reviewer is not familiar with the field of the paper and is not sure how novel the proposed method is and how difficult the problem of the paper is.

---

> ### Author Response · Authors · 2022-08-20
> **Author Response to Reviewer EzWb**
>
> Dear Reviewer EzWb,
>
> Thank you so much for your time in reviewing our paper, and thank you for your detailed feedback. In the following, we have addressed each comment in detail.
>
> ---
>
> **C1. Three important terms should be defined and explained clearly, which are 6 dof grasp, 7 dof grasp, and TSDF volume.**
>
> A1. All the acronyms are explained in the abstract and introduction sections.
>
> **C2. In Fig. 1, the color of the object should be different from one of the gripper for visibility.**
>
> A2. We have adjusted Figure 1 with the color of the object different with the gripper, and explicit annotations of surface point (red spheres) and grasp position (black spheres).
>
> **C3. In Fig. 2, it is difficult to see red points and green ones. Adding a part of zoom-in should help to see.**
>
> A3. We have zoomed in a part of the image to have a better visualization of the contact pairs.
>
> **C4. In Fig. 3, the color of 'g' should be the same as the line which represents a 'g' vector. The degree of transparency of the object should be high to see the vectors and points easily.**
>
> A4. We have extended the grasp axis and made it closer to ‘g’. The degree of transparency of the object has also been increased.
>
> **C5. In the problem definition, it should be clearer that the robot does not have the target object to grasp in the tasks of the paper and the robot selects the object which is easy to be grasped.**
>
> A5. We have modified the description of the problem to make it clearer.
>
> **C6. In the experimental section, a figure of a real robot in experiments should be added to make the fact that the method can be used in a real robot clearer.**
>
> A6. A figure of the real robot execution has been added to the revised manuscript.

---

> > ### Comment · Reviewer_EzWb · 2022-08-25
> > **A sentences of definition of a word should include this word**
> >
> > The explained 6 dof grasp, 7 dof grasp, and TSDF volume. However, the concrete definitions of these key words are still unclear for this reviewer. Adding a sentence of the definition of these words using these words can increase the clarity of the paper.

---

> > > ### Author Response · Authors · 2022-08-25
> > > **Author Response to Reviewer EzWb**
> > >
> > > The definitions of 6 dof grasp, 7 dof grasp, and TSDF volume are added to the introduction section in the revised paper.

---

> > ### Comment · Reviewer_EzWb · 2022-08-25
> > **The introduction should introduce a task**
> >
> > This reviewer found "In this work, we consider the scenario where a robot tries to pick any object out from the workspace one by one when all the objects are stacked together."  If the proposed method works only in the scenario, the introduction should explain the task definition because the grasping target object is given to a robot in other cases. The introduction should describe the problem which the paper tackles clearly.

---

> > > ### Author Response · Authors · 2022-08-25
> > > **Author Response to Reviewer EzWb**
> > >
> > > We have modified the description of the problem in L19-L22 to clarify that this paper tackles the problem of general object grasping [1] instead of instance grasping [2]. Moreover, we also replace the word “scenario” with “clutter removal task” in Sec 3.1 to make it clearer the setting of the problem. To have a consistent definition, we keep the comprehensive description of the problem in the Problem Definition section. Please refer to the latest revised paper to get more details.
> > >
> > > [1] Fang, Hao-Shu, et al. "Graspnet-1billion: A large-scale benchmark for general object grasping." Proceedings of the IEEE/CVF conference on computer vision and pattern recognition. 2020.
> > >
> > > [2] Cai, Junhao, et al. "Ccan: Constraint co-attention network for instance grasping." 2020 IEEE International Conference on Robotics and Automation (ICRA). IEEE, 2020.

---

### Official Review · Reviewer_cFdk · 2022-07-29

**Originality:** Good
**Technical Quality:** Very Good
**Clarity Of Presentation:** Very Good
**Impact:** 3

**Recommendation:**

Weak Accept: I recommend accepting the paper, but will not argue for my recommendation if the majority of other reviewers have a different opinion.

**Summary:**

This paper proposes a grasping-in-clutter framework for computing grasping contact pairs from a Truncated Signed Distance Function (TSDF). The paper argues that TSDFs capture more of the scene's geometry and the contact pair formulation removes the normal-based assumption present in other frameworks. From the TSDF, the Marching Cubes algorithm generates a mesh whose vertices serve as a grasp candidates. The framework next uses an antipodal grasping rule to find valid contact pairs, which are fed into a Contact-Point Network (CPN) that evaluates them for grasp quality. The CPN also takes the TSDF as input, allowing it to account for scene information. Finally, the gripper is collision checked along various approach vectors to find a valid one. To train the CPN, a large set of contact points are scored analytically and in simulation, based on whether they are graspability and collision-free. The method is compared against three other normal-based approaches, two of which use TSDFs.

**Issues:**

Connecting to what was stated above, it would strengthen the paper to clarify how the closed-loop pipeline is implemented. This seems particularly important given the emphasis of the closed-loop nature in the introduction.

Related to this, Sec 3.5 makes the comment that the modules "can" be implemented with GPU acceleration such that the framework can be efficiently executed closed-loop. If the experiments were done without this acceleration, can the process still be executed closed-loop? What aspects of the pipeline are a bottleneck?

Within the introduction, I didn't quite understand what meant by "first assumption may introduce position shift since most of the time the surface position is not exactly the same as the end-point position of the gripper,". Is this that the surface position is not the desired position of the gripper during the grasp? What is meant by position shift, is this during the grasping process? Perhaps clarifying the wording, or further using Fig. 1 as an example may help clarify this motivational assumption.

A few smaller comments, questions, notes:
- Making the annotations on Fig. 1 larger would make them easier to see.
- The appendix discusses that the step size of the Marching Cubes algorithm was set such that the number of vertices is around 1000-8000, which is "sufficient for covering the entire scene". How was this number chosen and how would changing this parameter impact the performance of the algorithm?
- Sec 5.1 mentions that any object that could not be grasped three times was manually removed. Where the type of objects that needed to be removed flat-shaped objects, as discussed in the limitations, or were there other classes of objects that the grasping systems struggled with?
- Given that the best performance is bolded in Table 1, it would maintain consistency to do a similar bolding in Table 2.


**Quality Of The Limitations Section:**

Limitations are addressed clearly

**Reviewer Expertise:**

3: The reviewer is fairly confident that the evaluation is correct

**Robotics Focus:**

Sufficient demonstration on hardware

**Strengths And Weaknesses:**

This paper focuses on a very relevant application area (grasping in clutter) and by using TSDFs the framework is able to factor in collision information about the scene. Removing the assumption to grasp normal to the object's surface could enable a wide range of grasping. The paper is well-organized and the related work highlights the motivation for the methodology. The experimental section compares to three state-of-the-art methods across a large set of simulation experiments and a clutter removal experiment. The discussion on how TSDF-based methods can struggle when grasping thin objects was particularly interesting.

The paper proposes a closed-loop grasping pipeline stating that as the robot moves the input from the sensor can be used to improve "the performance of the contact point detection". However it is not clear how the performance is improved over time. How is new information incorporated in? Would the grasping point ever change? Is the TSDF updated? Currently it is not clear how the closed-loop pipeline is implemented.

Particularly because two of the compared methods (VGN and VPN) use TSDF, it would be great to get further insight on how the proposed method uses the TSDF in a way that leads to improved performance. This could help tease out where the improvement comes from and how the aspects of the proposed approach connect back to the performance.

**Summary Of Recommendation:**

I am weakly recommending the acceptance of this paper. The paper builds on existing work to bring together several interesting ideas in order to enable a more generic form of grasping. The experimental section is fairly extensive and compares against several other relevant algorithms. The closed-loop pipeline is currently not clearly explained and there are a few clarifications (described below) that should be made.

---

> ### Author Response · Authors · 2022-08-20
> **Author Response to Reviewer cFdk**
>
> Dear Reviewer cFdk,
>
> Thank you so much for your time in reviewing our paper, and thank you for your detailed feedback. In the following, we have addressed each comment in detail.
>
> ---
>
> **C1. Connecting to what was stated above, it would strengthen the paper to clarify how the closed-loop pipeline is implemented. This seems particularly important given the emphasis of the closed-loop nature in the introduction.**
>
> A1. More details about the implementation of the closed-loop pipeline are included in Sec. 1.4 in the appendix. As suggested by reviewer AfSj, eHii, and EzWb, we replace Sec. 3.5 with a figure of real robot execution. And the original content is moved to Sec. 1.4 in the appendix.
>
> **C2. Related to this, Sec 3.5 makes the comment that the modules "can" be implemented with GPU acceleration such that the framework can be efficiently executed closed-loop. If the experiments were done without this acceleration, can the process still be executed closed-loop? What aspects of the pipeline are a bottleneck?**
>
> A2. Since TSDF fusion, CPN inference, and collision checking are all implemented with GPU acceleration, our method cannot achieve reactive grasping if these modules were executed without acceleration, and thus closed-loop grasping cannot be performed.
> Despite the advantages introduced by the TSDF volume, we believe it is also the bottleneck of the entire pipeline. Since the memory requirement for processing the volume scales cubically with the grid resolution, it may cost seconds when fusing a frame of depth map with only the CPU.
>
> **C3. Within the introduction, I didn't quite understand what meant by "first assumption may introduce position shift since most of the time the surface position is not exactly the same as the end-point position of the gripper,". Is this that the surface position is not the desired position of the gripper during the grasp? What is meant by position shift, is this during the grasping process? Perhaps clarifying the wording, or further using Fig. 1 as an example may help clarify this motivational assumption.**
>
> A3. We have adjusted Figure 1 with explicit annotations of surface point (red spheres) and grasp position (black spheres), and have elaborated on it in the introduction.
>
> **C4. The appendix discusses that the step size of the Marching Cubes algorithm was set such that the number of vertices is around 1000-8000, which is "sufficient for covering the entire scene". How was this number chosen and how would changing this parameter impact the performance of the algorithm?**
>
> A4. The number of samples is generated based on the complexity of the workspace and the step size of the Marching Cubes algorithm. For example, if the step size is set to 2 and there are 20 objects in the bin, the output mesh after the Marching Cubes algorithm will have a complex shape, and the number of vertices might be around 8000, according to our observation. In contrast, if the bin is empty, the output mesh is only a flat mesh, and the number of vertices will be less than 1000. Changing the step size will significantly affect the number of samples. We empirically set the step size to 2 to trade off the number of samples and the time efficiency.
>
> **C5. Sec 5.1 mentions that any object that could not be grasped three times was manually removed. Where the type of objects that needed to be removed flat-shaped objects, as discussed in the limitations, or were there other classes of objects that the grasping systems struggled with?**
>
> A5. Currently, we only found that small objects or flat-shaped objects are the hard samples for our methods.
>
> **C6. Some other minor issues**
>
> A6. Thank you for pointing out these issues. We have modified the table to make them more consistent.

---

### Official Review · Reviewer_eHii · 2022-07-31

**Originality:** Good
**Technical Quality:** Very Good
**Clarity Of Presentation:** Very Good
**Impact:** 3

**Recommendation:**

Strong Accept: I recommend accepting the paper and will argue for my recommendation even if other reviewers hold a different opinion.

**Summary:**

This paper proposes a grasp generation pipeline that reasons over a TSDF volume. To achieve this, the pipeline learns a grasp analysis model that reasons directly over meshes. This approach is validated over a wide range of experiments grasping objects in cluttered scenes. This pipeline significantly outperforms common baselines from normal-based grasp selection in both real-world and simulated experiments.

**Issues:**

My only issues with this paper lie in the presentation:

1. Figure 2. could be improved to better visually depict the proposed pipeline for unfamiliar readers.
2. Figure 4. can be confusing at first and is not well labeled.
3. The authors remark on the fact that their approach is trained exclusively on synthetic data, which is very impressive. However, this seems to be common among other pipelines as well. The reviewers should be careful with this statement.
4. The paper does not show any images of the real results or experimental setup. While not critical, it would be helpful to understand the difference between the synthetic training setup and the experimental setup.

**Quality Of The Limitations Section:**

Limitations are addressed clearly

**Reviewer Expertise:**

4: The reviewer is confident but not absolutely certain that the evaluation is correct

**Robotics Focus:**

Sufficient demonstration on hardware

**Strengths And Weaknesses:**

Strengths:

1. The paper is well written and overall easy to follow.
2. The new volumetric approach to grasping leads to major improvements in performance in comparison to normal-based approaches
3. This pipeline is validated over a large number of experiments, both real-world and simulated.
4. The paper is very clear and straightforward on its limitation and areas for improvement

Weaknesses:

1. The authors could improve the visual depiction of the pipeline for readers unfamiliar with the state-of-the-art grasping pipelines.

**Summary Of Recommendation:**

This paper presents a novel contribution to the field of grasping. The approach builds on top of previous work on volumetric grasping and demonstrates how this pipeline outperforms previous work on a wide array of experiments. The paper is well written, is technically correct, and a relevant contribution to the manipulation community. The reviewer recommends its acceptance.

---

> ### Author Response · Authors · 2022-08-20
> **Author Response to Reviewer eHii**
>
> Dear Reviewer eHii,
>
> Thank you so much for your time in reviewing our paper, and thank you for your detailed feedback. In the following, we have addressed each comment in detail.
>
> ---
>
> **C1. Figure 2. could be improved to better visually depict the proposed pipeline for unfamiliar readers. Figure 4. can be confusing at first and is not well labeled.**
>
> A1. In Figure 2, we zoom in on the part of the image to better visualize the contact pairs. For Figure 4, we remove the second subfigure, which might be redundant, and enlarge the others to have a clearer view.
>
> **C2. The authors remark on the fact that their approach is trained exclusively on synthetic data, which is very impressive. However, this seems to be common among other pipelines as well. The reviewers should be careful with this statement.**
>
> A2. We agree that current methods using simulated data only are common. Although we believe it is a novel way to generate contact labels in the TSDF volume, we do not consider it as our major contribution.
>
> **C3. The paper does not show any images of the real results or experimental setup. While not critical, it would be helpful to understand the difference between the synthetic training setup and the experimental setup.**
>
> A3. As pointed out by reviewer AfSj, Sec. 3.5 might be redundant. So, we move this section to the appendix and add a figure of the real robot in the revised manuscript.

---

### Official Review · Reviewer_AfSj · 2022-07-31

**Originality:** Fair
**Technical Quality:** Good
**Clarity Of Presentation:** Very Good
**Impact:** 1

**Recommendation:**

Weak Accept: I recommend accepting the paper, but will not argue for my recommendation if the majority of other reviewers have a different opinion.

**Summary:**

This paper proposes an algorithm for object grasping in clutter from a truncated signed distance function (TSDF) volume scene representation. The algorithm represents a grasp using a pair of points where the two fingers of a parallel jaw gripper are supposed to contact the object i.e. antipodal contact points. One of the pair of antipodal points is obtained by sampling from the input TSDF, and the other one by extending a vector backwards along the surface normal and intersecting it with the TSDF zero surface. Training is performed by generating data from a set of mesh objects arranged into cluttered scenes, labelled with theoretical antipodal score.

Experiments performed in simulation and on a real robot measure grasp success (GSR) and the success rate of clearing a clutter of objects (3 tries allowed per object to pick it up) (GCR).

**Issues:**

Ablation study experiments, especially to tease apart the effect of the proposed grasp representation.

**Quality Of The Limitations Section:**

Limitations are addressed clearly

**Reviewer Expertise:**

4: The reviewer is confident but not absolutely certain that the evaluation is correct

**Robotics Focus:**

Sufficient demonstration on hardware

**Strengths And Weaknesses:**

##  Strengths
- The paper is well written and understandable, figures are useful
- While this paper is not the first to introduce the idea of operating on a TSDF instead of a single image, the introduction section motivates the use of TSDF well
- Real robot experiments are conducted to verify the simulation results

## Weaknesses
- It is not clear how different this paper is from others that also operate on a (T)SDF volume, especially [12]
- One key difference is that [12] predicts 6-DoF gripper poses instead of contact point pairs as proposed here. However, the current experiments showing incremental performance improvement over [12] do not control for other factors, hence it cannot be said that the performance improvement is due to the grasp representation proposed here. For example, [12] uses an 8-layer network, while the network in this paper is more than twice deep (see appendix). EDIT: This was addressed in the authors' response below.

### Smaller points
- L15 "dexterous": this paper does not address dextrous manipulation
- L35-36: meaning unclear, difficult sentence to understand
- Section 3.5 is mostly redundant, the space can be used for other purposes e.g. images of real robot execution
- L189: "closed loop grasping pipeline": the pipeline described here is not closed loop - that would require the grasp sucess/failure signals influencing future grasps.
- L240: will the randomly paired points be antipodal? if not, why do this, because real world points will all be antipodal
- Quantitative comparisons to baselines in Table 3 are technically not comparable because they do not use the same test sets / scenes. This needs to be discussed in the paper if the current presentation is maintained.
- Supplementary material not referenced

**Summary Of Recommendation:**

I am voting to reject this paper because it is unclear whether the incremental performance improvement over [12] is because of the proposed grasp representation or a deeper network.
EDIT: Raising my rating to Weak Accept because the authors' response addresses my main concern.

---

> ### Author Response · Authors · 2022-08-20
> **Author Response to Reviewer AfSj**
>
> Dear Reviewer AfSj,
>
> Thank you so much for your time in reviewing our paper, and thank you for your detailed feedback. In the following, we have addressed each comment in detail.
>
> ---
>
> **C1. It is not clear how different this paper is from others that also operate on a (T)SDF volume, especially [12]**
>
> A1. The key differences between our work and [12] are 1) the representation of the gripper pose and 2) the procedure of sampling grasp candidates. In [12], the authors sample pose candidates by first considering the surface normal of each vertex as the approach axis of the gripper, and then rotating the gripper with multiple angles on the approach axis. Therefore, the orientation of all the candidates is restricted by the surface normal. On the contrary, we turn to consider the surface normal as the grasp axis, and incorporate it with the TSDF volume to retrieve another contact point. The final orientation of the gripper pose is determined by collision checking. As mentioned in L130-L131, the sampled candidates are more likely to be antipodal, which is also demonstrated in Table 1. Moreover, since we obtain the contact points, the exact position and the width of the gripper can be further computed, eliminating the position shift problem.
>
> **C2. One key difference is that [12] predicts 6-DoF gripper poses instead of contact point pairs as proposed here. However, the current experiments showing incremental performance improvement over [12] do not control for other factors, hence it cannot be said that the performance improvement is due to the grasp representation proposed here. For example, [12] uses an 8-layer network, while the network in this paper is more than twice deep (see appendix).**
>
> A2. According to Section III.C in [12], the authors mentioned that they use a twenty-layer 3D fully convolutional network as the backbone to extract features of the TSDF volume. Therefore, we actually use the same network structure, except the last layer is replaced by the one with only a single output value. The clarification is also included in L158-L160. As for the experiments, we use the same random seed when evaluating the performance of all the methods. Furthermore, we perform 1000 trials for each task to minimize the random error. Therefore, we believe the comparison is fair, and the results are statistically sufficient.
>
> **C3. L189: "closed loop grasping pipeline": the pipeline described here is not closed loop - that would require the grasp success/failure signals influencing future grasps.**
>
> A3. The closed-loop grasping in this paper means that we use the latest estimated pose as the visual feedback to correct errors between the current and estimated poses. Therefore, the trajectory of the arm will be generated on the fly based on the target pose estimated by the CPN.
>
> **C4. Quantitative comparisons to baselines in Table 3 are technically not comparable because they do not use the same test sets / scenes. This needs to be discussed in the paper if the current presentation is maintained.**
>
> A4. Since we only have two tables in the manuscript, we guess what you mentioned is Table 2. To have a fair comparison with the baselines, we set up the workspace with the same robot arm, background, vision sensor, and test objects. It is worth noting that the screwdriver in our experiment differs from the one tested by [12]. However, these two screwdrivers are all from the YCB object set and have a very similar shape. Thus, we believe it will not introduce much difference when performing the grasp experiments. We will clarify the difference in the appendix.
>
> **C5. L240: will the randomly paired points be antipodal? if not, why do this, because real world points will all be antipodal**
>
> A5. To train the contact point network, we need both positive and negative samples. And L238-240 describe 3 different ways to generate the negative samples. Randomly pairing points can be regarded as an augmentation trick to increase the negative samples.
>
> **C6. Some other minor issues**
>
> A6. Thank you for pointing out these issues. We have modified the description to make it clearer and more proper in the revised manuscript.

---

### Author Response · Authors · 2022-08-20
**Volumetric-based Contact Point Detection for 7-DoF Grasping**

**Comment:**

We thank all reviewers for their constructive feedback and for helping us make our paper more organized!

We have revised our manuscript to address reviewers’ comments. The major changes are listed below:

Section 1: Adjust the color of the objects in Figure 1 and add more annotations to illustrate the difference between grasp position and surface point.

Section 1: Elaborate more on the position shift problem in L26-L34.

Section 1: Zoom in a part of contact labels in Figure 2 to have better visualization.

Section 3: Change the degree of transparency of the object and extend the grasp vector in Figure 3.

Section 3: Move Sec. 3.5 of the paper to Sec. 1.4 of the appendix.

Section 5: Bold the results with the best performance in Table 2.

Section 5: Add a figure of real robot experiment.

Appendix: More details about the implementation of the closed-loop pipeline are included in Sec. 1.4.

The revised paper and the appendix are attached in this comment for your convenience. Hopefully we have addressed all your concerns and questions.

**Zip File:**

/attachment/d65e5d4d0109bf5f51520c5bed1a8b57d67347dc.zip

---

> ### Author Response · Authors · 2022-08-25
> **Revised PDF**
>
> Updated 25th, Aug, 2022
>
> We have updated our revised paper according to the latest comments from the reviewer. All the changes are highlighted in blue in the pdf.

---

### Meta-Review · Area_Chair_VtK2 · 2022-08-14

**Recommendation:** Accept (Poster)
**Confidence:** 4

**Metareview:**

This paper proposes a 7DoF grasping method which is based on TSDF for scene representation to compute grasp contact pairs, where the Marching Cubes algorithm generates a mesh whose vertices serve as a grasp candidates. These contact pairs, TSDF, vertex normals and surface vertices are used as input for a proposed a Contact-Point Network (CPN) to compute grasp quality. The paper is well written and easy to follow. The use of TSDF for an application of grasping in clutter to extract collision information about the scene is sensible. Experiment results show improvements. There are real robot experiments to verify the simulation results.
The authors have clarified most concerns during the rebuttal phase. Though there are other issues, e.g. it would also be helpful to know if changing the grasp pose at each close-loop execution step ever causes discontinuous or undesirable behavior, I would like to recommend the paper can be accepted.

---

> ### Author Response · Authors · 2022-08-25
> **Author Response to Area Chair VtK2**
>
> Dear Area Chair VtK2,
>
> We sincerely thank you for the comprehensive summary of reviews from all perspectives and the suggestions for improving the paper.
>
> As for the issues and concerns raised, we have carefully responded to each reviewer and revised the manuscript correspondingly. Concretely, we have addressed the main concerns by:
>
> illustrating the critical difference between our method and the VPN proposed by [12] (AfSj),
>
> clarifying that the performance boost attributes to the representation of the pose instead of a deeper neural network (AfSj), and
>
> providing more details about the implementation of the closed-loop pipeline (cFdk).
>
> Correspondingly, we have revised the paper with the major changes listed in the above comments and highlighted in blue in the pdf.
>
> Please check the detailed reply for more information and feedback.